# Determination of Munsell Soil Colour Using Smartphones

**DOI:** 10.3390/s23063181

**Published:** 2023-03-16

**Authors:** Sadia Sabrin Nodi, Manoranjan Paul, Nathan Robinson, Liang Wang, Sabih ur Rehman

**Affiliations:** 1School of Computing, Mathematics and Engineering, Charles Sturt University, Bathurst, NSW 2795, Australia; 2Centre for eResearch and Digital Innovation, Federation University, Mount Helen, VIC 3350, Australia; 3Global Centre for Environmental Remediation, The University of Newcastle, Callaghan, NSW 2308, Australia; 4School of Computing, Mathematics and Engineering, Charles Sturt University, Port Macquarie, NSW 2444, Australia

**Keywords:** soil colour, Munsell colour, NIX pro sensor, smartphone

## Abstract

Soil colour is one of the most important factors in agriculture for monitoring soil health and determining its properties. For this purpose, Munsell soil colour charts are widely used by archaeologists, scientists, and farmers. The process of determining soil colour from the chart is subjective and error-prone. In this study, we used popular smartphones to capture soil colours from images in the Munsell Soil Colour Book (MSCB) to determine the colour digitally. These captured soil colours are then compared with the true colour determined using a commonly used sensor (Nix Pro-2). We have observed that there are colour reading discrepancies between smartphone and Nix Pro-provided readings. To address this issue, we investigated different colour models and finally introduced a colour-intensity relationship between the images captured by Nix Pro and smartphones by exploring different distance functions. Thus, the aim of this study is to determine the Munsell soil colour accurately from the MSCB by adjusting the pixel intensity of the smartphone-captured images. Without any adjustment when the accuracy of individual Munsell soil colour determination is only 9% for the top 5 predictions, the accuracy of the proposed method is 74%, which is significant.

## 1. Introduction

Soil colour is a key indicator of soil physical and chemical properties such as mineral composition, organic matter, and moisture content, and it is an important feature used in soil classification [1]. During field research of soil, scientists frequently carry a soil colour book, because colour is typically employed to detect morphological properties [2]. The Munsell Soil Colour Book (MSCB) has been used in US soil surveys to reduce personal idiosyncrasy since around 1949 [3]. It is the most commonly used book for soil colour identification, and for this research project, the 2009 edition of MSCB has been used. The Munsell system is described by three variables: hue, value and chroma. Hue indicates shade, value indicates lightness, and chroma indicates saturation [3]. Soil colour determination using the soil colour charts has been undertaken for over 50 years by experienced soil and land surveyors in Australia. These observations of hue, value and chroma of the matrix colour often occur in field conditions using the MSCB. The method and procedure are detailed in the Australian Soil and Land Survey Field Handbook [4]. The use of cameras and phones has been problematic in the past in field conditions, therefore assessments of colour have been undertaken by trained soil and land surveyors. Neurobiological research shows that the reflection spectra of colour chips in Munsell colour charts match the sensitivity of cells of human eyes [5]. Therefore, the MSCB has been used for many decades for colour specification to determine soil properties and soil organic materials.

The traditional method of soil colour determination involves mapping the human perception to the colour chips of the MSCB [6]. The process of using human perception to determine soil colours does not guarantee an accurate determination, as it differs from person to person and varying external conditions such as light and time of the day, as highlighted by [7,8,9,10,11].

Soil colour is commonly determined by soil scientists and archaeologists under natural lighting conditions, but most of the studies in this area have been done in a controlled environment with controlled illumination conditions [12], or required external sensors [13]. In recent years, we have seen a revolution in the quality of smartphones with in-built good-quality cameras. Even some are almost equivalent to professional cameras, and we can see the growing use of smartphones in different domains [14,15]. In [12], the authors presented a comparison between smartphones and visual soil colour determination that shows that the smartphone performs better. This study highlights that smartphones provide one of the most efficient and cost-effective methods for soil colour classification.

However, almost all research in this area depends on external light or relevant instruments, and is conducted in a controlled environment. In a study conducted by [16], the authors have proposed a smartphone-based soil classification sensor, CMOS (Complementary Metal Oxide Semiconductor), which can be used as an attachment kit and includes a calibration card, shading bucket, and external lenses. In similar studies, [17,18], the authors have designed and presented a web application to recognize Munsell soil colour for archaeologists. They presented a demonstration application of their experiments based on their experimental results using a digital camera in controlled experimental settings. However, in field research, having a controlled experimental setup is not convenient, and procuring a digital camera can prove to be costlier than a smartphone itself.

As previously discussed, observing soil colour using MSCB is highly influenced by observant expertise, colour vision, and experience. Earlier in [19], the authors discussed the accuracy of visually determining Munsell soil colour based on field and weather conditions. Many subsequent studies have therefore looked at and proposed alternative ways to measure soil colour accurately. One of the most promising and commonly used technologies used today is smartphones, which contain built-in high-quality camera sensors, Global Positioning System (GPS) sensors, ambient light sensors, etc. The newer models emerging in the market have more developed camera sensors and image technologies, which makes the device a promising option for colour specification. Some researchers in [20] reviewed a series of research cases that used smartphone cameras in the field of agriculture, environment and food. They showed that there is huge potential for smartphones and their cameras in the agricultural industry. Table 1 shows the list of research that has been undertaken recently in the food, plants, water, and soil domains using smartphone cameras. Similarly, a study by [21] estimated soil structure, texture, and drainage, and indicated a good approach for estimating soil health and fertility. This suggests that smartphones and their cameras have great potential for colour detection. Although a smartphone camera has a high potential for colour detection, there are many potential confounding factors that should be considered, such as different camera sensors, lighting conditions, weather, colour calibration, etc. For that reason, research has been undertaken mostly in controlled environments using digital cameras and external sensors. Research by [22] was carried out on topsoil in Scotland using a Fuji digital camera to demonstrate calibration models for several soil variables (carbon, pH, nitrogen, etc.). This study was also done in a controlled environment, by maintaining constant distance, angle, and light levels.

In this research, our priority is to determine soil colours without the use of external sensors in order to provide a cost-effective and convenient solution to stakeholders. In the initial phase of this research, we used two commonly available smartphones: the Samsung S10 and Google Pixel5 under indirect sunlight to capture soil colours as per MSCB. We used these two smartphones because, at the time of data collection, both scored very well for their camera and image quality [23,24]. The primary camera specification of Samsung S10 is a 12 MP telephoto lens (45°) [25] and Google Pixel5 is 12.2 MP [23]. In the next phase of this study, we compare the images captured via smartphones with the true colour determined using a Nix Pro-2; a low-cost sensor used for colour-matching across various industries. The Nix colour sensor has been used for rapid quantitative prediction of soil colour. It has been used to define soil colour [6,13,26,27] and predict soil organic properties [28,29]. As the current literature claims that the Nix Pro [30] colour sensor offers highly accurate colour discrimination, we opted to use this sensor as our source of ground-truth data, and compared it with the images captured by a smartphone. We used the Nix Pro to verify if our system was working or not. Once we completed developing the system, the Nix Pro was of no further use. The relationship between colour acquired from images of Munsell soil colour chips and Nix Pro-based determination has not been compared in previous studies.

In this research, we found a discrepancy between smartphone-captured images and the actual colour acquired from the Nix Pro. There is an almost constant weight difference between the two data sets and the difference varies for different smartphones. Figure 1 shows the colour discrepancies in the three colour components (Red, Green, Blue) of the RGB colour model. The focus of this research is to determine the relationship based on the discrepancies so that we can adjust the smartphone-captured colour readings to obtain a true colour match. Obviously, there are many attributes to consider, such as time of the day, lighting conditions, different colour models, different smartphones, different distance functions, and geographical location to obtain more accurate soil colour profiling using a smartphone. We have also implemented a ranking-based method to determine Munsell soil colour using two colour difference models. We have ranked the closest determined colours and compared them with the Nix Pro-generated colours. If a colour is matched at rank 1, it means the true colour has been determined on the first prediction. Thus, we have calculated the accuracy of our proposed colour determination method.

The overall objective of this study was to investigate the accuracy and prediction rate of soil colour classification under indirect sunlight in natural outdoor conditions from mobile captured images. The key contributions of this work are summarised as follows:Analyse the colour discrepancy between images captured by smartphones and the Nix Pro colour sensor;Propose a novel approach to accurately capture soil colour, irrespective of the capturing method for a specific geographic area; andFind the most suitable colour model and corresponding distance function by investigating different colour models and colour-matching distance functions.

The rest of the article is organised as follows: Section 2 gives information on the materials and methods used in this study. Section 3 consists of all the results and analysis that have been done in this study, and lastly, we conclude this article with Section 4.

## 2. Materials and Methodology

### 2.1. Munsell Soil Colour Book

The most commonly used soil colour references, which have been used since 1949 are the Munsell soil colour charts [3]. This chart, or the MSCB, is widely used by professional soil scientists for soil judging. The book consists of coloured papers or chips mounted on the hue cards, showing multiple variations of value and chroma in vertical and horizontal directions [10]. Users match the closest colour between their soil samples and the Munsell soil colour chips. We are using the 2009 edition of the MSCB for this study, which consists of 443 colour chips and 13 hue cards: 5R, 7.5R, 10R, 2.5YR, 5YR, 7.5YR, 10YR, 2.5Y, 5Y, 10Y–5GY, GLEY1, GLLEY2, and WHITE PAGE. For instance, the 5R hue card is shown in Figure 2.

The Munsell colour system is a colour space that is based on three properties: hue (Basic Colour), chroma (colour intensity), and value (lightness) also known as HVC, which can be represented cylindrically in three dimensions as shown in Figure 3. The hue is divided into five dominants: red, yellow, blue, green, and purple, and is subdivided between 0 and 10, commonly in steps of 2.5. Value is represented by the spine of hue which indicates the lightness of colour along a scale 0–10 (darkest to lightest) [47]. Chroma is represented as the purity of colour, with lower chroma being less pure and is measured vertically outward from the vertical axis.

### 2.2. Nix Pro Colour Sensor

Nix Pro has been used in many studies previously and has proven to be closely accurate to determine colours in various domains, such as soil and agriculture, paint and print, and the food industry [6,28]. Soil scientists also reported the usefulness of the Nix Pro colour sensor as it provides a reasonably accurate colour prediction [26]. Research [13] shows that the Nix Pro colour sensor provides true soil colour regardless of the moisture condition of the soil, enabling it to be used both on dry and wet soil. For this reason, we have employed the Nix Pro 2 to gather our ground-truth data to compare that with our image-generated data to determine the accuracy of our proposed method, as highlighted in Figure 4.

The overall approach is to determine the Munsell soil colour using a smartphone, and to do that we have compared 2 different colour models and identified which model is best to work with. To do that, we have compared the colour difference (Euclidean distance, CIE1976 and CIE2000) of each chip with the Nix Pro-generated data. Smartphones produce standard red-green-blue (sRGB) colour space where these 3 colours are primary colours, and this production is device-dependent. We have compared RGB colour spaces with device-independent colour space International Commission on Illumination or CIELAB. CIELAB colour space is a three-dimensional colour space and covers the entire human colour perception, where L indicates lightness, A colour in red and green, and B colour in yellow and blue.

For the RGB colour space, we have used Euclidean distance, which is the standard means of determining distance. For CIELAB-based colour difference, we have used and analysed CIE1976 [48] and CIE2000 [49] to determine the closest colour. The standard means of determining the distance between two colours (linear dimension) in an RGB colour model is Euclidean distance, as it is basically a straight distance between two points. There are several distance functions available, such as CIE1976, CIE1994, CIE2000, CMC l:c (1984), and so on. However, not all the colour models work based on that principle. CIE1976 was formulated to measure colour differences for CIELAB colour coordinates. It is similar to Euclidean distance but for the CIELAB colour model. On the other hand, CIE2000 is a more recent scheme that covers more corrections to determine the colour difference. The authors in [50,51] have mentioned the following three corrections:Hue rotation term to mitigate the problem of blue region;Compensate for neutral colours;Compensate for lightness, chroma, and hue.

These computations do some adjustments for better results. Figure 5 shows the entire process that has been followed to determine the closest Munsell soil colour. After converting RGB to LAB value, we have calculated the colour difference for each chip with all 443 Nix Pro-generated data. Lowest colour difference refers to the closest determined colour. Theoretically, the lowest difference colour match should match with the Nix Pro-generated match, but the result shows that it matches after the 50th prediction. In our proposed method, the prediction accuracy increases to a great extent.

### 2.3. Data Collection and Pre-Processing

The data-collection process involves two steps. First, we have acquired our ground-truth data as shown in Figure 4. The Nix Pro 2 generates data for different colour models, such as RGB, LAB, CMYK etc. We took the RGB and CIELAB values from the Nix Pro, each of the 443 chips of the MSCB (2009 edition). In the next step, we captured images of each page 13 of the MSCB using a Samsung S10 and Google Pixel5 under different environmental conditions. When capturing the images, we maintained an indirect sunlight condition by maintaining a shadow on the book to eliminate unnecessary reflection of sunlight. We have followed a free-style image acquisition method so the distance between the book and the smartphone is not constant. Generally, when an end user takes a photo, they use the auto settings of the camera app. In the proposed method, we used the auto camera setup available in smartphones. There are several data-processing steps behind the camera software when the auto camera setup is used. The auto settings automatically do image processing according to various things such as lighting conditions, focus, colour calibration, etc. We did not perform an analysis of the sensitivity of the processing software. Using a Samsung S10, we collected images from each hour from 9 a.m. to 5 p.m. on 20 March 2022 (9 sets) and using a Google Pixel5 we collected five sets of data from 10 a.m. to 3 p.m. on different days in January 2022. As we have implemented a free-style data-collection process, data pre-processing was needed before using the images. We used Photoshop to crop each of the 443 soil colour chips by 150 px/150 px and saved them as separate data sets. We also measured the ambient illuminance with the Samsung S10 to determine the best time of the day for data collection. We used a lux meter app that captures illuminance using the device’s ambient light sensor.

### 2.4. Determining the Closest Munsell Soil Colour and Rank

First, we calculated the average RGB of a particular colour chip and then converted it to the CIELAB colour space. The colour difference using CIE1976 and CIE200 colour difference models of each chip with all 443 data generated from Nix Pro were calculated. The lower colour distance means the closer the colours, which means the lowest distanced colour is the best prediction. However, that does not happen all the time. To check colour determination, we ranked all the determined colours from lowest to highest and identified in which rank the Nix Pro-determined data are being matched (Table 2 and Table 3). Rank 1 means that exact determination has been found in the first prediction and rank 443 means the worst prediction. We can see that with the Samsung S10 and using Euclidean distance and the RGB colour model, this gives us the worst average ranking. The CIE1976 and CIE2000, in contrast with the CIELAB colour model, achieve a much better ranking. For that reason, we have only focused on CIELAB for further analysis.

### 2.5. Average Colour Difference

From our analysis, we saw that there is a pattern between Nix Pro data and image data in almost all the chips. We calculated average weight differences for CIELAB values for both the Samsung S10 and Google Pixel5. For the Samsung S10, the LAB weight difference is 12.89, 3.14, and 5.06 and for the Google Pixel5 the differences are 8.06, 3.02, and 4.49, as shown in Table 4. We then added this weight difference to the same image data and predicted again and found that the rank went to 13 from 49 for the Samsung S10 and to 16 from 29 for the Google Pixel5 as shown in Table 2 and Table 3. This clearly highlights a better prediction rate. The phone’s processor performance does affect the data results, but for this study, we did not focus on the internal processor individually. We only used the smartphones to capture images, and noticed that for these two smartphones, the pixel performances are different, and, to obtain better accuracy, we need different kinds of adjustments as shown in Table 4.

### 2.6. Employ Location-Based Prediction

The MSCB has 13 different hues, but not all the colour is prominent in all parts of the world. Soil colour and characteristics depend on the environment, and for that reason, the colour of the soil is different in different places. Therefore, not all 13 hues are relevant or existent across all places. The colour of soil depends on organic matter, moisture content, and other physical and chemical properties. A study by [52] was carried out collating over 680,000 observations on Australian topsoil had 166, 7000 soil colour observations with the most common hues, including 5Y,2.5Y,10YR,7.5YR,5YR,2.5YR and 10R. For this study, we have employed a location-based prediction and only predicted the colour chips of the previously mentioned 7 hues as mentioned in Table 5 and Table 6.

## 3. Results

### 3.1. Better-Performing Colour Model

Measurement of soil colour is usually done using MSCB, which follows the Munsell HVC system, but the system is not uniform and relies on user perception and comparison [53]. There are other colour space models available, such as RGB, CIELAB, CIEXYZ, and CIELUV. As smartphone cameras produce RGB images directly, we have used this colour model as part of our comparison. We have compared the RGB colour model with the CIELAB colour model as it is device-independent. We have used Euclidean distance to rank RGB-based prediction and CIE1976, CIE2000 distance function for the CIELAB colour model. From Table 2 and Table 3 we can see that the average rank of predicting Munsell soil colour using RGB is as follows: for the Samsung S10 it is 89 and for the Google Pixel5 it is 56. For the CIELAB colour model, the prediction rate increased for both the Samsung S10 and Google Pixel5 with average prediction rank 49 (CIE2000) and 29 (CIE2000), respectively. Therefore, we can clearly prove that CIELAB is the better-performing colour model for Munsell soil colour prediction.

### 3.2. Best Time of the Day to Capture Images Using a Smartphone Camera

To determine which time of the day or which sunlight condition is best for capturing soil images using a smartphone camera, we have gathered 9 sets of data using the Samsung S10 from 9 a.m. to 5 p.m. each hour. Additionally, lighting conditions or illuminance was measured under both generally and under indirect sunlight. In general, the illuminance gives us an idea on the weather conditions of the day at different times and the lux of indirect sunlight is the shadowed condition under which the soil images were taken, as shown in Table 7. The table clearly shows that from 10 a.m. to 4 p.m., the prediction rate is similar, and we can see a rise in rank in the early morning (9 a.m.) and late afternoon (5 p.m.). That means between 35,000 lx and 76,000 lx outside is preferable for capturing images of soil. Therefore, any time during the day except early morning and evening is a good time to capture images using a smartphone camera for soil colour prediction.

### 3.3. Added Weight and Location-Based Prediction

In Figure 1, we can see a constant pattern between the Nix Pro-generated data and the image-generated data for both employed smartphone cameras. We calculated the average difference between image-generated value and Nix Pro-generated value of L, A, and B of the CIELAB colour model in Table 4. We added the calculated weight difference with our image-produced data and saw that the prediction rate improved significantly, as shown in Table 2 and Table 3. For he Samsung S10, the average prediction rank went from 48 to 12 and for the Google Pixel5 from 28.93 to 16 with the CIE2000 distance method.

The location-based prediction model, where we only focused on 7 particular hues which are prominent in Australian agricultural soil, achieved an even better result. When we combined both added weight and the location-based method, we determined that the Samsung S10 average prediction rank is approximately 8, as shown in Table 5 and for Google Pixel5 it is approximately 7 as shown in Table 6 for both CIE colour difference methods. For both smartphones, on average the exact match can be found under 10 predictions. This is considerably better than the normal prediction, which is ranked greater than 30.

### 3.4. Distance Function

For this study, we used 3 distance functions for Munsell soil colour prediction, namely Euclidean distance for the RGB colour model and CIE1976 and CIE2000 for the CIELAB colour model. As the RGB colour model did not perform better than CIELAB, this left the CIE1976 and CIE2000 distance functions for evaluation. However, for both smartphone images, CIE1976 and CIE2000 performed similarly, as highlighted in Figure 6 and Figure 7. We can see that the curves for the distance functions overlap with each other. The Google Pixel5 gives the best prediction, with added weight and a location-based method, with an average prediction rank of 6th when CIE1976 was used, whereas Samsung S10 gives the second-best prediction with an average prediction rank of 8th when CIE1976 was used. The difference between the two distance functions is negligible, as shown in Table 8.

Our proposed method can determine the Munsell soil colour chips by taking images from the book. In addition, it is related to actual soil colour. The ultimate goal of this study is to determine the actual soil colour. At this initial stage, we verified our method using Nix Pro. In addition, the advantage is that we are currently using smartphone-captured images to identify the Munsell soil colour chips and the results are showing promising results. There are a couple of challenges to using actual soil colour. There are 443 Munsell soil colour chips in the book. However, because of various ingredients in soil, soil can be any colour. In future studies, we aim to quantify with the real soils and identify which is the closest Munsell soil colour. The key contributions are summarised as follows:Between RGB and CIELAB colour model, CIELAB performs much better.CIE1976 and CIE2000 distance functions perform similarly for both the Samsung S10 and Google Pixel5The best time to capture images to determine Munsell soil colour is between 10 a.m. and 4 p.m. (35,000 lux to 76,000 lux). The prediction rate decreased in the early morning and evening times.According to our data and analysis between the Samsung S10 and Google Pixel5, the Google Pixel5 performed better for Munsell soil colour prediction (Figure 8).The best prediction rate was achieved when we employed our proposed weight and location-based prediction model. After adding weight and focusing on soil colours in Australia, 70% of the time, the exact match was found in the top 5 prediction when we employed Samsung S10. The Google Pixel5 (Table 9) was able to match the colour in the top 5 predictions 74% of the time. However, we performed our study on sunny days, and the results may vary on cloudy and rainy days. To achieve similar determinations under different weather conditions, further research and investigations are necessary.

## 4. Conclusions

In this paper, we have evaluated the performance of two widely used smartphone cameras for determining Munsell soil colour at different times of the day, using three distance functions. We have also proposed a colour-intensity relationship between Nix Pro and smartphone images and the proposed weight increased the Munsell soil colour prediction rate. Predicting the colours directly from the images does not give an accurate result and the prediction rate is very low. Additionally, as different smartphones have different kinds of camera sensors, processing performance varies a lot. To minimize these issues, several adjustments are needed for soil colour prediction. For that reason, we did an analysis to find a suitable colour model and distance functions to work with. Through experiments, we have found a pattern between the captured images and the actual colour acquired from the Nix Pro. There is an almost constant weight difference between the two datasets and the difference varies for different smartphones. The accuracy of predicting the exact colour chip on top 5 prediction using the Samsung S10 and Google Pixel5 is 7% and 9%, respectively, for CIE1976. This increased considerably when we balanced the weight difference, and the prediction rate went up to 52% for the Samsung S10 and 34% for the Google Pixel5. The results give an even higher soil colour prediction rate when we only focused on possible soil colours found in Australia. Overall, the Google Pixel5 gave the best result by balancing the weight difference and focusing on specifically Australian soil colours with 74% accuracy rate on top 5 predictions. However, we performed our study on sunny days, and the results may vary on cloudy, rainy or any other days. To achieve similar determinations under different weather conditions, further research and investigations are necessary. In addition, we have used auto settings for the camera app in this study, and the result may vary with some specific settings. The sensitivity and prediction error of the proposed model also needs to be analysed. In future research, we aim to analyse our method with real soil and with more smartphones and their different settings under various environmental conditions, to make a robust and accurate identification.

## Figures and Tables

**Figure 1 sensors-23-03181-f001:**
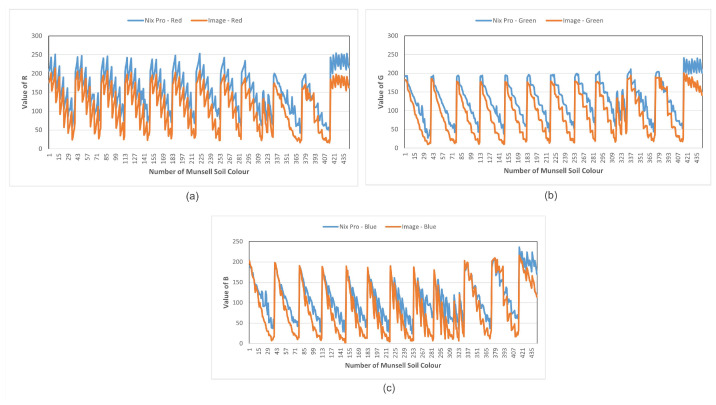
Colour discrepancies between the Nix Pro and Samsung smartphone provided pixel intensity of RGB colour model. The colour components are (**a**) Red colour, (**b**) Green colour, and (**c**) Blue colour.

**Figure 2 sensors-23-03181-f002:**
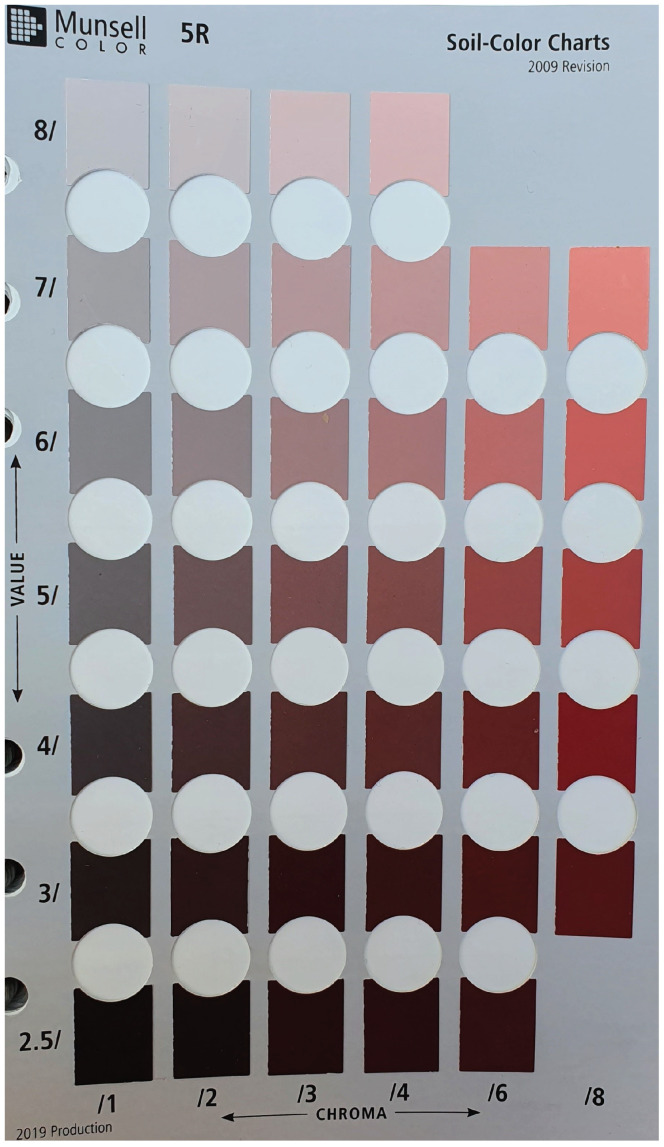
Example of the 5R card of Munsell Soil Colour Book.

**Figure 3 sensors-23-03181-f003:**
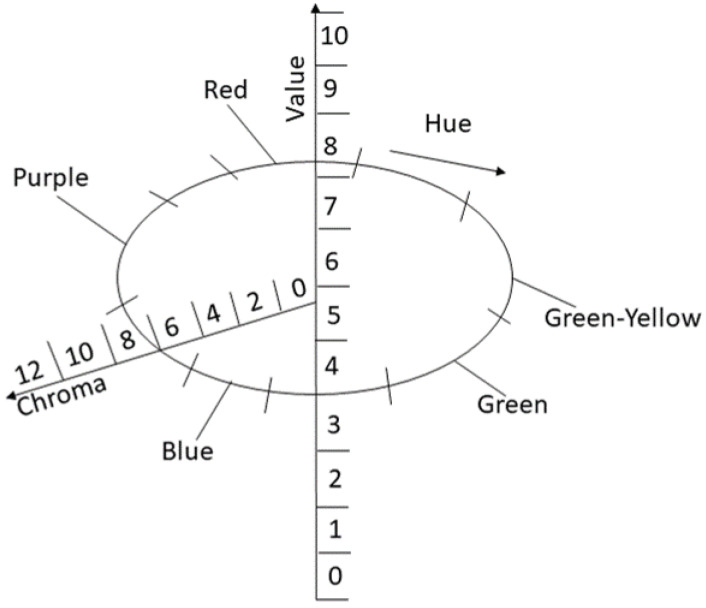
Munsell colour system.

**Figure 4 sensors-23-03181-f004:**
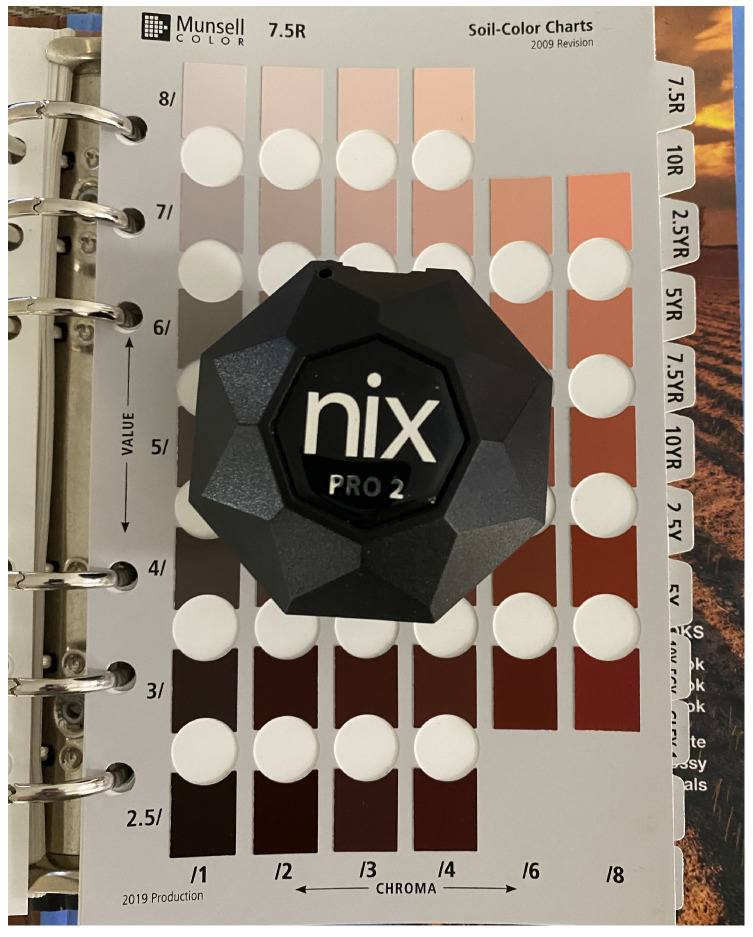
Data collection of ground-truth data using Nix Pro.

**Figure 5 sensors-23-03181-f005:**
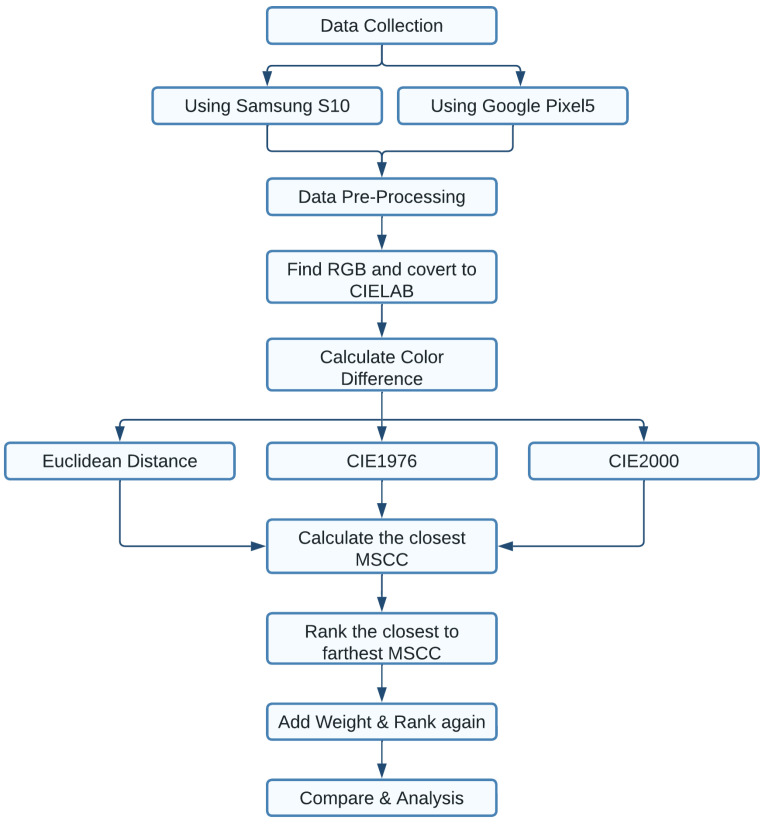
Flow diagram of the process.

**Figure 6 sensors-23-03181-f006:**
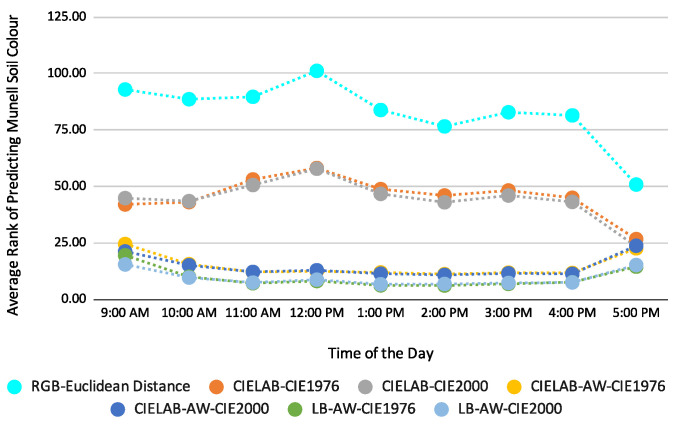
Average prediction rank using Samsung S10; Colour models used: RGB and CIELAB; Distance Functions employed: Euclidean distance, CIE1976, CIE2000; LB—Australian Location-Based, AW—added weight.

**Figure 7 sensors-23-03181-f007:**
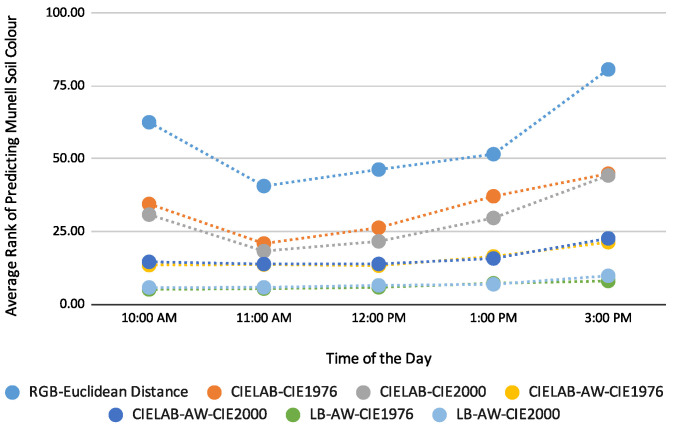
Average prediction rank using Google Pixel5; Colour models used: RGB and CIELAB; Distance Functions employed: Euclidean distance, CIE1976, CIE2000; LB—Australian Location-Based, AW—added weight.

**Figure 8 sensors-23-03181-f008:**
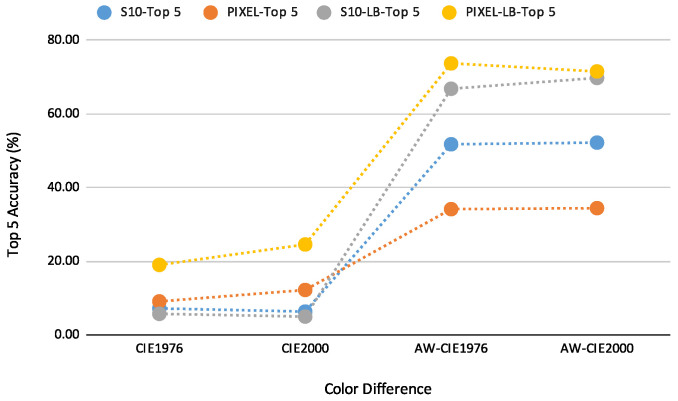
Accuracy curve for top 5 prediction accuracy for Samsung S10 and Google Pixel5. Colour models used: RGB and CIELAB; Distance Functions employed: Euclidean distance, CIE1976, CIE2000; LB—Australian Location-Based, AW—added weight.

**Table 1 sensors-23-03181-t001:** List of research using smartphone cameras.

No.	Application Area	Publication
1.	Detect harmful substance in milk	[31]
2.	Detect alkaline phosphate	[32]
3.	Methanol determination in sugar cane	[33]
4.	Quantify salmonella	[34]
5.	Detect microbial contamination	[35]
6.	Determine fat	[36]
7.	Determine crop seed	[37]
8.	Measure foliar damage	[38]
9.	Estimate chlorophyll	[39]
10.	Identify weed	[40]
11.	Colorimetric reader	[41]
12.	Detect biochemicals	[42]
13.	Measure water salinity	[43]
14.	Chlorine monitor	[44]
15.	Water quality monitor	[45]
16.	Soil type classification	[16]
17.	Soil water erosion	[46]

**Table 2 sensors-23-03181-t002:** Samsung Galaxy S10 average prediction rank using RGB and CIELAB colour model and three distance functions—Euclidean distance, CIE1976, and CIE2000. AW—Proposed added weight difference from Table 4.

Set	Time	RGB-Euclidean Distance	CIELAB-CIE1976	CIELAB-CIE2000	CIELAB- AW-CIE1976	CIELAB- AW-CIE2000
Set 2	10:00 a.m.	88.59	43.01	43.44	15.50	15.02
Set 3	11:00 a.m.	89.66	53.03	50.58	12.02	12.08
Set 4	12:00 p.m.	101.16	58.12	57.80	12.37	12.83
Set 5	1:00 p.m.	83.81	48.76	46.66	11.86	11.30
Set 7	3:00 p.m.	82.81	48.16	45.92	11.77	11.42
AVERAGE		89.21	50.22	48.88	12.70	12.53

**Table 3 sensors-23-03181-t003:** Google Pixel average prediction rank using RGB and CIELAB colour model and three distance functions—Euclidean distance, CIE1976, and CIE2000. AW—Proposed added weight difference from Table 4.

Set	Time	RGB-Euclidean Distance	CIELAB-CIE1976	CIELAB-CIE2000	CIELAB- AW-CIE1976	CIELAB- AW-CIE2000
Set 1	10:00 a.m.	62.53	34.47	30.81	13.49	14.60
Set 2	11:00 a.m.	40.62	20.89	18.29	13.67	13.85
Set 3	12:00 p.m.	46.29	26.32	21.63	13.28	13.88
Set 4	1:00 p.m.	51.56	37.12	29.66	16.45	15.74
Set 5	3:00 p.m.	80.67	44.86	44.25	21.34	22.64
AVERAGE		56.33	32.73	28.93	15.65	16.14

**Table 4 sensors-23-03181-t004:** Average weight difference between Nix Pro data and image data.

	L	A	B
Samsung S10	12.89	3.14	5.06
Google Pixel 5	8.06	3.02	4.49

**Table 5 sensors-23-03181-t005:** Samsung S10 average prediction rank using CIELAB colour model and CIE1976 and CIE2000 distance functions. Location-Based—Prediction using only prominent soil colours in Australian topsoils. AW—Proposed weight difference from Table 4.

Set	TIME	Location-Based Focused Hue-CIE1976	Location-Based CIE2000	Location-Based AW-CIE1976	Location-Based AW-CIE2000
Set 2	10:00 a.m.	28.08	33.41	9.92	9.58
Set 3	11:00 a.m.	35.55	38.95	7.06	7.37
Set 4	12:00 p.m.	39.42	44.57	8.04	8.64
Set 5	1:00 p.m.	30.47	33.10	6.12	6.62
Set 7	3:00 p.m.	29.82	32.70	6.78	7.18
AVERAGE		32.76	36.55	7.58	7.88

**Table 6 sensors-23-03181-t006:** Google Pixel average prediction rank using CIELAB colour model and CIE1976 and CIE2000 distance functions. Location-Based—Prediction using only prominent soil colours in Australian topsoils. AW—Proposed weight difference from Table 4.

Set	TIME	Location-Based Focused Hue-CIE1976	Location-Based CIE2000	Location-Based AW-CIE1976	Location-Based AW-CIE2000
Set 1	10:00 a.m.	17.86	17.08	5.13	5.80
Set 2	11:00 a.m.	13.83	13.58	5.40	5.84
Set 3	12:00 p.m.	13.65	11.97	5.85	6.52
Set 4	1:00 p.m.	25.03	20.66	7.24	6.89
Set 5	3:00 p.m.	28.04	30.12	8.03	9.81
AVERAGE		19.68	18.68	6.33	6.97

**Table 7 sensors-23-03181-t007:** Luminous intensity captured from 9 a.m. to 5 p.m. using Samsung S10 smartphone under direct sun light and indirect sunlight (shadow on the book). In addition, average prediction rank using CIELAB colour model and CIE1976 and CIE2000 distance functions. Location-Based—Prediction using only prominent soil colours in Australian topsoils. AW—Proposed weight difference from Table 4.

Set	TIME	Weather Condition (lx)	Indirect Sun Light Intensity (lx)	CIELAB- AW-CIE1976	CIELAB- AW-CIE2000	Location-Based AW-CIE1976	Location-Based AW-CIE2000
Set 1	**9:00 a.m.**	11,982	3098	**24.43**	**21.16**	**19.43**	**15.39**
Set 2	10:00 a.m.	35,510	2673	15.5	15.02	9.92	9.58
Set 3	11:00 a.m.	46,329	3432	12.02	12.08	7.06	7.37
Set 4	12:00 p.m.	75,952	3380	12.37	12.83	8.04	8.64
Set 5	1:00 p.m.	74,220	3054	11.86	11.3	6.12	6.62
Set 6	2:00 p.m.	75,856	2993	11.09	10.73	6.05	6.73
Set 7	3:00 p.m.	59,037	3300	11.77	11.42	6.78	7.18
Set 8	4:00 p.m.	38,204	3180	11.63	11.22	7.51	7.43
Set 9	**5:00 p.m.**	13,715	1675	**22.51**	**23.68**	**14.41**	**15.12**

**Table 8 sensors-23-03181-t008:** Summary of average prediction rank for both Samsung S10 and Google Pixel5 smartphones using CIELAB colour model and two distance functions (CIE1976 and CIE2000). Location-Based—Prediction using only prominent soil colours in Australian topsoils. AW—Proposed weight difference from Table 4.

Colour Difference	Samsung *S*10	Google *Pixel*5	Samsung *S*10-Location-Based	Google *Pixel*5-Location-Based
CIE1976	50.22	32.73	32.76	19.68
CIE2000	48.88	28.93	36.55	18.68
AW-CIE1976	12.70	15.65	7.58	6.33
AW-CIE2000	12.53	16.14	7.88	6.97

**Table 9 sensors-23-03181-t009:** Accuracy in percentage in top 5 prediction for both Samsung S10 and Google Pixel5 smartphones using CIELAB colour model and two distance functions(CIE1976 and CIE2000). Location-Based—Prediction using only prominent soil colours in Australian topsoils. AW—Proposed weight difference from Table 4.

Colour Difference	Samsung *S*10 TOP 5 Prediction Accuracy (%)	Samsung *S*10 Location-Based TOP 5 Prediction Accuracy (%)	Google *PIXEL*5 TOP 5 Prediction Accuracy (%)	Google *PIXEL*5 Location-Based TOP 5 Prediction Accuracy (%)
CIE1976	7.18	5.71	9.07	18.99
CIE2000	6.32	4.96	12.19	24.54
AW-CIE1976	51.74	66.81	34.13	**73.70**
AW-CIE2000	52.19	**69.75**	34.40	71.51

## Data Availability

Not applicable.

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
