# Peer review of "Determination of Munsell Soil Colour Using Smartphones"

_sensors, 2023, doi:10.3390/s23063181_

Round 1
Reviewer 1 Report
Manuscript Number: sensors-2242712
Title: Determination of Munsell Soil Colour Using Smartphones
For: Sensors
Overview:
In this work, the authors attempted to use different smartphones to determine the Munsell color of the soil for application with soil monitoring and data analysis. In addition, this work analyzes the color differences between images captured by smartphones and the Nix Pro color sensor, and finds the most suitable color model and the corresponding distance function. Overall, the research methodology is feasible and the study objectives and results are innovative and appeal to a relevant readership. However, after a careful assessment of the manuscript, the reviewer found that there were a few errors in this manuscript. The research work is innovative, but the paper has some unclear logical expressions, and the analysis and discussion of the results are not deep enough. Therefore, I recommend a major revision so that the manuscript needs to be improved in quality and readability.
General Comments:
1. Please consider the relevance of this work to the journal Sensors. If the authors did not use the Nix Pro color sensor, but only a smartphone, is the pixel capture and analysis system for smartphones a relevant study for sensors?
Specific Comments:
Title section
2. Good.
Abstract section
3. The research's importance is very well presented. However, the scientific question in the abstract is presented in a very large number of words, please streamline it. Also, please use more descriptions for the presentation of the result.
4. In line 13, what is the proposed method? Please specify.
Introduction section
5. Line 54, no extra sensors used, does that include not using the Nix Pro color sensor?
6. The color differences in Figures 1-3 can be combined into one figure because they attempt to illustrate the same scientific problem.
7. Line 54, "In this research" and line 67, "In our research", have these studies been reported? Please consider using this research foundation to highlight the focus of this work.
"Related work" section
8. Please consider if this section is duplicated with the introduction section. Do the authors consider only the differences in graphics algorithms? But smartphones also have very different processing performances for images, does this have an impact on the results?
"Materials" and " Methodology " section
9. I suggest that these two parts might be written together.
10. The method of photographing the Australian soil color should be explained, and it is better to provide a comparison figure comparing the real image color.
11. The related distance function should also be provided
"Results" section
12. Are Samsung S10 and Google Pixel5 two different image processing algorithms? Please provide clarification. And does the phone processor performance affect the data results?
13. Regarding your proposed method and conventional image processing software, please explain the sensitivity of soil color data processing and error analysis in the prediction process.
14. The analysis here does not go far enough, the authors do not analyze the errors from the perspective of a mathematical model, and are the predictions an evaluation of the actual soil color data? Please consider a deeper analysis to convince the reader of your proposed method.
15. Figures 8-10 only illustrate that these methods differ in soil color identification and do not clearly illustrate the advantages of your proposed approach. Please consider adding real analysis examples, for example in soil classification. If possible, please add a discussion section devoted to the advantages of your proposed solution.
"Conclusions " section
16. The writing is not very standardized, usually, the conclusion section is a summary of the findings, and no graphs or figures appear. Instead, Figure 10 and Table 9 are suggested to be added to the analysis of results and discussion section.
References section
17. Please double-check all literature in the full paper to avoid errors.
Reviewer 2 Report
The paper “Determination of Munsell Soil Colour Using Smartphones” is connected with modern colorimetry. The results of soil color measurements by using two smartphones are presented and discussed. The text is understandable, and the presentation quality is good. This paper is both attractive and valid. However, I have recognized some issues that must be improved before publication. Therefore, I recommend a “major revision,” and I encourage the Authors to make the proper corrections.
DETAILED COMMENTS:
1. Figures (e.g., 1,2, and 3) and tables (e.g., 3 and 4) must be located near the relevant description.
2. Figure 5 must be placed near the description of the Munsell System in the introduction.
3. The explanation of why these two particular smartphones were chosen for this experimental study is required. Moreover, the detailed technical data sheet of sensors used in them must be added.
4. L.197 An explanation of how and for what purpose photoshop was used is needed. The exemplary figure presenting analyzed samples is necessary.
L.202 What does “light intensity” mean, and how was it measured? The improper terminology must be eliminated (lux values -> illuminance). The same remark for L.262. It should be “That means between 35000 lx and 76000 lx…”
5. L.257 “The Lux of direct sunlight…” This sentence is not true. Please remove or replace it with an adequate explanation of the weather conditions.
6. Much more insightful comparison of the results received for the different weather conditions must be added.
8. Fig. 8 and 9. The obtained data presented in these figures seem discrete, not continuous. Therefore these graphs seem to look different (-> trend line).
7. Description of sections (e.g., L.127, L.157-158, L.238-239) are unnecessary. Please remove it.
9. Authors claim that they present only initial results. Therefore, further research proposals must be indicated in conclusion.
10. The conclusion section needs to emphasize the general outcome. It must be added.
Round 2
Reviewer 1 Report
Manuscript Number: sensors-2242712_v2
Title: Determination of Munsell Soil Colour Using Smartphones
For: Sensors
Overview:
Thank you very much for your serious consideration of my comments, and for having given reasonable modifications. I have seen a satisfactory response. I think the current version is basically in line with the journal's publication requirements, and I look forward to more exciting work from you in the future. However, there are some details or errors that need to be checked again
Specific Comments:
1. Please double-check any possible hidden mistakes in the manuscript before publication to avoid regrets. I hope the manuscript will be published perfectly.
2. Although the authors answered all my questions with what they thought were perfect answers, the sensitivity and prediction error of this model still need to be considered in future studies.
Reviewer 2 Report
I have familiarized with the improved version of this paper. Thank you very much for prepared the precise corrections and answers. Right now, I recommend this article for publication. Congratulations and all the best.
